# The Neurovascular Unit as a Locus of Injury in Low-Level Blast-Induced Neurotrauma

**DOI:** 10.3390/ijms25021150

**Published:** 2024-01-17

**Authors:** Gregory A. Elder, Miguel A. Gama Sosa, Rita De Gasperi, Georgina Perez Garcia, Gissel M. Perez, Rania Abutarboush, Usmah Kawoos, Carolyn W. Zhu, William G. M. Janssen, James R. Stone, Patrick R. Hof, David G. Cook, Stephen T. Ahlers

**Affiliations:** 1Neurology Service, James J. Peters Department of Veterans Affairs Medical Center, 130 West Kingsbridge Road, Bronx, NY 10468, USA; 2Department of Neurology, Icahn School of Medicine at Mount Sinai, One Gustave Levy Place, New York, NY 10029, USA; georgina.perez-garcia@mssm.edu; 3Department of Psychiatry, Icahn School of Medicine at Mount Sinai, One Gustave Levy Place, New York, NY 10029, USA; miguel.gama-sosa@mssm.edu (M.A.G.S.); rita.de-gasperi@mssm.edu (R.D.G.); 4Mount Sinai Alzheimer’s Disease Research Center and the Ronald M. Loeb Center for Alzheimer’s Disease, Icahn School of Medicine at Mount Sinai, New York, NY 10029, USA; carolyn.zhu@mssm.edu (C.W.Z.); patrick.hof@mssm.edu (P.R.H.); 5General Medical Research Service, James J. Peters Department of Veterans Affairs Medical Center, Bronx, NY 10468, USA; 6Research and Development Service, James J. Peters Department of Veterans Affairs Medical Center, 130 West Kingsbridge Road, Bronx, NY 10468, USA; gissel05@gmail.com; 7Department of Neurotrauma, Operational and Undersea Medicine Directorate, Naval Medical ResearchCommand, 503 Robert Grant Avenue, Silver Spring, MD 20910, USA; rania.abutarboush.ctr@health.mil (R.A.); usmah.kawoos.ctr@health.mil (U.K.); stephen.t.ahlers.civ@health.mil (S.T.A.); 8The Henry M. Jackson Foundation for the Advancement of Military Medicine Inc., Bethesda, MD 20817, USA; 9Department of Geriatrics and Palliative Care, Icahn School of Medicine at Mount Sinai, New York, NY 10029, USA; 10Nash Family Department of Neuroscience, Icahn School of Medicine at Mount Sinai, New York, NY 10029, USA; bill.janssen@mssm.edu; 11Friedman Brain Institute, Icahn School of Medicine at Mount Sinai, New York, NY 10029, USA; 12Department of Radiology and Medical Imaging, University of Virginia, 480 Ray C Hunt Drive, Charlottesville, VA 22903, USA; jrs7r@virginia.edu; 13Geriatric Research Education and Clinical Center, VA Puget Sound Health Care System, 1660 S Columbian Way, Seattle, WA 98108, USA; dgcook@uw.edu; 14Department of Medicine, University of Washington, 1959 NE Pacific St., Seattle, WA 98195, USA

**Keywords:** animal models, astrocytes, blast, inflammation, traumatic brain injury, vascular pathology

## Abstract

Blast-induced neurotrauma has received much attention over the past decade. Vascular injury occurs early following blast exposure. Indeed, in animal models that approximate human mild traumatic brain injury or subclinical blast exposure, vascular pathology can occur in the presence of a normal neuropil, suggesting that the vasculature is particularly vulnerable. Brain endothelial cells and their supporting glial and neuronal elements constitute a neurovascular unit (NVU). Blast injury disrupts gliovascular and neurovascular connections in addition to damaging endothelial cells, basal laminae, smooth muscle cells, and pericytes as well as causing extracellular matrix reorganization. Perivascular pathology becomes associated with phospho-tau accumulation and chronic perivascular inflammation. Disruption of the NVU should impact activity-dependent regulation of cerebral blood flow, blood–brain barrier permeability, and glymphatic flow. Here, we review work in an animal model of low-level blast injury that we have been studying for over a decade. We review work supporting the NVU as a locus of low-level blast injury. We integrate our findings with those from other laboratories studying similar models that collectively suggest that damage to astrocytes and other perivascular cells as well as chronic immune activation play a role in the persistent neurobehavioral changes that follow blast injury.

## 1. Blast-Induced Neurotrauma in the Military

Public awareness of traumatic brain injury (TBI) in the military greatly increased over the past decade because of the conflicts in Iraq and Afghanistan [1] where estimates are that 10–20% of returning veterans suffered a TBI [2,3]. While military-related TBIs occur for various reasons, certain types of TBIs are relatively unique to military settings, the most prominent being blast-induced neurotrauma (BINT). For service members in Iraq and Afghanistan, exposure to improvised explosive devices (IEDs) caused most TBIs [2,3,4,5].

Initially, focus in the most recent conflicts was on the moderate-to-severe end of the TBI spectrum [6], the type of injury that would be recognized in-theater, and the conflict in Iraq led to the highest number of service-related severe TBIs since the Vietnam era [7]. However, as hostilities continued, what became apparent was that many returning veterans were appearing at Department of Veterans Affairs (VA) medical facilities with symptoms suggesting residual effects of mild TBIs (mTBIs) never recognized during service. Indeed, mTBIs greatly outnumbered moderate-to-severe TBIs during these conflicts [2,3].

In addition to in-theatre exposures, there is increasing concern regarding the effects of subclinical blast exposure [8,9,10]. This type of exposure, now being referred to as military occupational blast exposure, is common for many service members during training and military operations [8]. Whether this type of repetitive, low-level blast exposure causes health problems later in life is unclear [8,11], but recent studies suggest that cumulative low-level blast exposure over a service member’s career is associated with chronically worse brain health, more post-traumatic stress disorder (PTSD)-related symptoms, and a heightened risk for developing symptoms after a later blast injury [12,13,14,15]. Studies in breachers who routinely use explosives in their operations to, e.g., gain entry into buildings, further show chronic structural and functional brain imaging changes as well as altered blood biomarkers suggestive of brain injury [16]. Recent news media reports have brought additional attention to this topic [17].

A history of TBI is frequent among veterans seeking treatment at VA mental health facilities [18]. TBI has been linked to numerous mental health problems including depression, anxiety, poor impulse control, sleep disorders, and suicide [19,20]. A striking feature in the most recent veterans has been the frequent association of blast-related TBI with PTSD [1,21]. The association of BINT and mental health problems is not new and has historical roots dating back to the entity known as shell shock first recognized during World War I [22].

How much of the association of blast-related TBI with mental health problems is driven by blast-related physical trauma vs. psychologically based trauma needs more study [1,21]. However, PTSD and depression are the most prominent drivers of the worsening functional status seen in many veterans who suffered concussive blast injuries in Afghanistan [23]. Supporting a role for blast-related mechanisms, rats exposed to repetitive, low-level blast exposure develop a variety of chronic PTSD-related behavioral traits that are present more than one year after injury [24,25,26,27,28,29].

Over the last 15 years, interest in how blast exposure affects the nervous system has led to a rapid expansion of clinical as well as animal studies [21,30,31]. Here, we review findings from a model of low-level blast exposure in rats that we have been studying for more than 10 years that exhibits an early and selective vascular injury [24,25,26,27,28,29,32,33,34,35,36,37,38,39,40,41,42]. We focus on the neurovascular unit (NVU) as a locus of initial BINT injury and how changes in the perivascular environment might lead to the chronic neurobehavioral disturbances seen in this model in the absence of initial direct neuronal injury. We integrate findings from our own studies with those from others who have been studying the effects of low-level BINT.

## 2. A Rat Model of Low-Level Blast Injury with Chronic PTSD-Related Behavioral Traits

In 2012, Ahlers et al. [43] described an animal model of BINT developed to mimic a level of blast exposure that would be associated with human mild TBI or subclinical exposure. Initial studies established that exposures up to 74.5 kPa (equivalent to 10.8 psi) caused no post-exposure apnea or mortality [43]. While representing a level of blast transmitted to brain [44], these exposures produced only mild transient behavioral disturbances and no widespread brain histopathology [24,43]. Examination of the lungs showed no hemorrhagic petechiae or other pathology often caused by blast exposure [43]. By contrast, higher exposures (120 kPa, 17.4 psi) led to frank subdural and intraparenchymal hemorrhages and visibly evident histopathology in brain along with pulmonary hemorrhages, effects more consistent with moderate-to-severe BINT in the setting of polytrauma [43].

For most studies in our subsequent work, blast exposures were delivered to male rats at 10 weeks of age, with rats lying prone in line with the long axis of the shock tube and the head closest to the blast tube driver which generates the shock wave. Head motion was restricted during exposure to minimize damage from rotational/acceleration injury, which is in keeping with the limited head movements expected of humans exposed to comparable blast forces [45,46]. To mimic the multiple blast exposures commonly experienced by service members in Iraq and Afghanistan [47], for most studies described in this report, rats were subjected to 74.5 kPa exposures (impulse 175.8 kPa * ms; duration 4.8 ms) [43] delivered once per day for 3 consecutive days (3 × 74.5 kPa).

Rats subjected to 3 × 74.5 kPa repetitive low-level blast exposure develop cognitive and PTSD-related behavioral traits including anxiety, enhanced acoustic startle, impaired recognition memory, and exaggerated fear learning [24,25,26,27]. These traits develop in a delayed manner, being absent in the first eight weeks after blast exposure but consistently present three to four months and longer after exposure. Once established, traits remain present for more than one year after blast exposure [24,25,26,27] and are likely present for the lifetime of the animal. These animals thus model the chronic neurobehavioral syndromes that veterans often suffer following BINT [23,48,49].

## 3. The Neurovascular Unit

The brain receives the largest blood supply of any organ [50]. However, unlike other organs, CNS blood supply is heavily filtered with substances that enter or leave the CNS regulated [51,52]. Regulation is performed at the level of the blood–brain barrier (BBB) in the context of a complex NVU whose cellular elements include an intraluminal glycocalyx, endothelial cells, perivascular astrocytes, microglia, other mural cells including pericytes, and smooth muscle cells as well as perivascular nerves [51,52]. An extracellular matrix is associated with these structural elements that gives rise to inner and outer basement membranes or basal laminae. An inner basement membrane surrounds the endothelial cells, separating them from the most closely associated cells, the pericytes, while an outer basement membrane lies between the pericytes and the astrocytic endfeet [53,54]. Within the blood vessel lumen exists a complex glycocalyx that additionally regulates blood vessel properties [55].

The NVU regulates cerebral blood flow as well as transport of glucose and other metabolites into the CNS [52,56]. It also controls the movement of solutes and clears metabolic waste out of the CNS [52,56]. Collectively, the NVU is critical for homeostatic maintenance and protection of the CNS from toxins or other biologically active molecules, some of which, while functionally beneficial in the periphery, may disrupt CNS homeostasis.

For blood-borne substances to enter the CNS, they must first transit vascular endothelial cells and their associated basement membranes [56]. The CNS vascular endothelium differs from that found in most peripheral organs by the absence of fenestrations and the presence of tight junctions between endothelial cells, which limit paracellular passage of water-soluble molecules that would otherwise enter most peripheral organs [57]. This matrix of neurovascular endothelial cells, closely knit together by their tight junctions, comprises the main elements of the blood–brain barrier (BBB) that governs what can and cannot cross into the brain [58]. CNS endothelial cells also exhibit less pinocytotic activity and the passage of many substances occurs only through active transport mechanisms specific to individual substrates [57]. The basement membranes, which help to maintain the proper association of elements within the NVU, also serve as a reservoir for growth factors and adhesion proteins that regulate BBB permeability [57,59].

Astrocytes interact with blood vessels through their endfeet, where they directly contact endothelial cells and mural cells [60]. Most of the brain capillary network is generated postnatally [61], at the same time that the astrocytic endfeet start to engage the vasculature [62]. Astrocytic endfeet cover nearly the entire surface of the brain’s blood vessels and contain specialized organelles and local protein translation machinery. Astrocytic endfeet have their own unique molecular composition, including scaffold proteins, anchor channels, transporters, receptors, and enzymes that regulate how astrocytes interact with blood vessels [60]. Astrocytic endfeet are thought to help regulate BBB function and govern regional cerebral blood flow to match metabolic supply and demand via microvascular dilation and constriction [60,63]. By their contribution to perivascular structure, astrocytes also regulate the glymphatic clearance system, which transports solutes out of the brain [64,65].

Pericytes are considered multi-functional cells embedded within the walls of capillaries and post-capillary venules [66]. Various functions have been attributed to them, including regulating cerebral blood flow and BBB integrity [66]. They have a role in vascular development and, in adults, behave like stem cells [66]. Pericytes also regulate neuroinflammatory states and immune cell entry into the CNS [66]. Microglia further contact endothelial cells, astrocytes, and neurons, where they regulate neuroinflammatory states as well as modify synaptic contacts [67,68,69].

Smooth muscle cells are the most abundant cell type found in the vascular wall [70]. Smooth muscle is found on all arteries, arterioles, and veins, but is absent from capillaries and small post-capillary venules. The lymphatics also have a smooth muscle layer. Vascular smooth muscle cells contract or relax to regulate intravascular blood volume and local blood pressure [70]. Smooth muscle contraction also regulates periarterial drainage of soluble metabolites [71].

Finally, neurons, through perivascular nerves, make contact with endothelial cells, astrocytes, and, likely, other mural cells, allowing neuronal activity to regulate vascular tone through effects on endothelial cells, smooth muscle, and astrocytes [52]. Thus, a variety of cellular elements within the NVU could be disturbed by blast exposure and contribute to brain dysfunction after BINT.

## 4. The Vasculature as a Locus of Injury in Low-Level BINT

Blast injury has long been known to affect the cerebral vasculature [31]. Acutely, higher-level blast exposures of 120 kPa or above show a prominent hemorrhagic component in species including rats, mice, rabbits, ferrets, and goats [43,72,73,74,75,76,77]. Varying degrees of subdural, subarachnoid, and intracerebral hemorrhage are visible grossly and microscopically along with edema and vascular congestion on the brain surface. Effects seem similar whether exposures are under shock tube conditions or exposure to live explosives, with a correlation between injury severity and blast overpressure levels or distance from the detonation of live explosives. While higher-level blast exposures have a prominent vascular component, their association with contusions and intraparenchymal and subdural hemorrhage as well as, likely, direct shock wave damage to neuronal and glial elements, make them a mixture of vascular and non-vascular injuries.

In the model that we have been studying, rats exposed to single or multiple (three) 74.5 kPa blast exposures develop a selective vascular pathology in the absence of a more generalized histopathology. General neuropathological screens, including histology (hematoxylin and eosin/Nissl) and a variety of immunostains, do not reveal any consistent histo- or immunopathology [24,32,42]. These exposures can produce some degree of hemorrhage in the choroid plexus and lateral ventricles. Also observed is a type of likely shear-related lesions, which typically follow the course of penetrating cortical vessels [42]. These lesions may be unique to BINT, but because of their focal nature and infrequency, their contribution to the overall pathological effects is not clear.

In multiple studies using electron microscopy (EM) [34,35,37,38,42], we have noted a generalized microvascular pathology in the setting of an otherwise normal neuropil (Figure 1). What is notable in Figure 1 is that despite the abnormalities in blast-exposed vessels, no abnormalities are present in the neighboring neuropil except at the very margins of severely affected blood vessels. While it is difficult to exclude that there may not be some subtle neuronal injury that is not apparent by standard histology or immunostaining, the striking difference between the vascular pathology and the surrounding neuropil suggests that the brunt of the physical blast injury is on the cerebral vasculature.

## 5. Comparisons to Other Models of Low-Level BINT

To compare our results with others, one must first address what low-level BINT is in an animal model. As noted above, initial studies in our model established an overpressure exposure up to 74.5 kPa as compatible with a low-level BINT [21,43]. This consideration was based on these exposure conditions, producing a blast wave that is transmitted to the brain [44], but which does not produce major gross neuropathological or systemic effects nor long-lasting behavioral effects in the immediate recovery period [24,43]. The lack of evidence for coup/contrecoup injuries further supports the relatively mild nature of the brain injury [24,43]. Thus, we later suggested 74.5 kPa as a dividing line between a low-level blast compatible with a low-level BINT as might occur in mTBI or subclinical blast exposure as opposed to moderate-to-severe TBI associated with more widespread tissue damage including subdural and intraparenchymal hemorrhage [21]. Others who have examined this question have come to relatively similar conclusions concerning the magnitude of an overpressure wave consistent with low-level blast exposure [46].

The general requirements for BINT modeling have been discussed in other reviews [45,78,79,80]. Notably, factors other than maximal overpressure of the blast wave contribute to the tissue-damaging potential of a blast wave. The duration and shape of the waveform (i.e., the area under the curve referred to as the impulse) correlate better with tissue damage than raw overpressure. Indeed, pressures as high as 550 kPa generated by live explosives and delivered over brief 0.2 ms durations cause no visible neuronal, axonal, or vascular pathology, as judged by standard histological and immunohistochemical stains [81]. By contrast, 120 kPa exposures delivered in a pressure wave of several ms, which is on the order of the durations produced by most shock tubes, produce extensive tissue damage [43].

Other factors which affect the tissue-damaging potential of a shock wave are body shielding and head restraint. Indeed, it seems that the use of some form of body shielding or selective head exposure has become the norm in many laboratories doing blast research. Body shielding clearly affects CNS vascular injury and is discussed below in the section on direct endothelial injury. Head restraint also affects tissue injury in that the lack of muscle tone in an anesthetized rodent allows for head oscillations and accelerations, creating what has been referred to as a bobblehead effect. These oscillations create the potential for significant rotation/acceleration injuries independent of the effects of the primary blast wave, effects that have been clearly demonstrated by studies in mice [82]. Additionally, some groups expose only one side of the head, which likely increases the potential for a bobblehead effect. While rotation/acceleration injuries from head oscillations are part of most moderate-to-severe blast-related TBIs in humans, they are likely relatively minor factors in subclinical blast exposures and in most blast-related mTBIs.

Despite the rapid expansion of blast research over the last decade, low-level BINT is still relatively little studied. Table 1 summarizes rodent models of blast exposure that have utilized blast overpressures of 80 kPa or less. The studies are hard to compare directly as the use of body shielding and head restraint have been variable. Many studies have described prominent vascular pathology, although some have not. However, those that have not did not examine the vascular ultrastructure with EM. Other studies have described extra-vascular pathology, including neuronal pathology, but it is hard from the published descriptions to judge the relative prominence of vascular vs. non-vascular pathology.

In a mouse model, perhaps the most comparable to ours in rats, Gu and colleagues have studied open-field blasts in mice using a single live explosion [86,87,88,89]. Their protocol generates a 46.6 kPa peak overpressure with an impulse of 60 kPa * ms, and a primary wave duration of 3 ms, parameters somewhat lower than those used in our rat blast model. They report that single-blast injuries do not result in extensive tissue damage at the light microscopy level in brain, although they identified ultrastructural abnormalities in myelin sheaths, mitochondria, and synapses [87,88,89]. In their most recent study [86], multiple ultrastructural changes were described in the NVU, including changes in endothelial cells, pericytes, and basement membranes, as well as swelling of astrocytic endfeet, that seem similar to those seen in our rat model (described below). Clearly, future studies in multiple models seem warranted to clarify the relative selectivity of the vasculature to low-level BINT.

## 6. Is Direct Damage to Endothelial Cells by BINT a Thoracic Effect?

The endothelial cell is the initial contact point for substances entering the CNS. Direct endothelial cell damage could affect BBB properties as well as disrupt endothelial cell connections to mural cells, the vascular basement membrane, and perivascular astrocytes. Disruption by BINT could thus affect multiple elements of the NVU, contributing to the functional consequences discussed in other sections.

As illustrated in Figure 1, the endothelial cell is damaged by low-level BINT. A question of interest is how this damage occurs. The primary blast wave may affect the brain as it is transmitted through the tissue. BINT may also be associated with acceleration/rotation of the head, imparting mechanical energy to the brain that may cause injuries similar to those seen in non-blast TBI, although this is probably less of an issue with low-level BINT if the head is restrained. A third mechanism that has been suggested is that the blast wave striking the body causes indirect CNS injury through what has been referred to as a thoracic or systemic mechanism [95,96]. Current thinking on this last mechanism conceptualizes the kinetic energy of the shock wave as being transferred into hydraulic energy that is carried through the systemic vasculature causing rapid blood displacement into the lower-pressure intracranial compartment [96,97,98,99]. Such a transmission of energy would be predicted to preferentially damage cellular elements close to cerebral vessels.

The importance of this mechanism has been indirectly assessed by using selective brain exposure or body shielding. While some groups such as ours deliver whole-body exposures, many investigators routinely subject only the head to blast exposure or use various forms of shielding to protect the body. While there are few direct comparisons in the same laboratory, in reviewing studies across many labs, it seems clear that a given blast overpressure exposure produces less intracranial pathology if delivered as a head-only exposure compared to a whole-body exposure (reviewed in [31]). In addition, higher head-only exposures seem necessary to produce the same pathology as lower whole-body exposures [31]. In the few studies that compared head-only vs. whole-body blast exposures in the same experimental system [96,100,101], all found that pathology was significantly less with a head-only exposure.

In a study that addressed this mechanism perhaps the most directly, Simard et al. [102] delivered selective blast exposures to the thorax or jugular veins of rats. They showed acute perivenular inflammatory changes induced by thorax-only exposure that were prevented by ligating the jugular vein and reproduced by selective jugular vein exposure [102]. Taken along with the multiple other studies showing that shielding can reduce or eliminate BINT effects on the brain [96,100,101], the studies of Simard et al. [102] make a compelling case for direct endothelial cell and NVU damage by transmission of a pressure wave through the vasculature.

One study has tried to address the thoracic mechanism in humans [103]. In this study, special weapons and tactics (SWAT) personnel undergoing breacher training were fitted with a jugular vein compression collar. Interestingly, the collar mitigated fMRI (functional magnetic resonance imaging) changes on a working memory task performed during training [103]. More study of this phenomenon seems warranted.

## 7. Blast Damage to the Endothelial Glycocalyx

The glycocalyx is a carbohydrate-rich layer that coats the luminal surfaces of vascular endothelium [55,104]. It imparts a negative electric charge to endothelial cell surfaces, forming an electrical barrier. Endothelial glycocalyx differs in the CNS from many peripheral tissues in being found over the entire luminal surface of cerebral capillaries while it only partially covers capillaries in organs such as heart and lungs [55]. The glycocalyx in the CNS is also more resistant to stripping from lipopolysaccharide-induced vascular injury, suggesting that CNS glycocalyx has a denser structure [55]. The glycocalyx is thought to play roles in endothelial protection, regulating BBB properties, leukocyte adhesion to endothelial cells and nitric oxide production and as a layer of the vasculature responsible for sensing luminal shear forces [104,105,106,107,108]. Glycocalyx degradation causes BBB dysfunction [105]. The endothelial glycocalyx is thus thought crucial for maintaining brain homeostasis.

While we have not analyzed the glycocalyx in our model of 3 × 74.5 kPa blast exposure, Hall et al. [109,110] studied how repeated low-intensity blast affects the brain capillary glycocalyx using 12 × 40 kPa blast exposures conducted with a minimum of 24 h between blasts. They found that blast exposure reduced the glycocalyx length and density in multiple brain regions. A functional significance to glycocalyx damage was suggested by the observation that administering hyaluronidase, an enzyme that binds to and degrades hyaluronan (a major glycocalyx structural component), prior to blast exposure reduced deficits in water maze performance and induced a thickening of the glycocalyx.

Another study by Chen et al., (this Special Issue) explored the longitudinal effects of a single high-intensity blast (130 kPa, 18.9 PSI) exposure on degradation and recovery of the of glycocalyx, alterations in cerebral blood flow, and the responsiveness of cerebral blood flow to hypercapnia. A significant reduction in the density of the glycocalyx was observed 2–3 h, 1 day, and 3 days after blast exposure. Recovery of the glycocalyx was observed at 28 days post-blast along with an increase in the components of the glycocalyx (heparan sulphate at 14 and 28 days and syndecan-2 and chondroitin sulphate at 28 days post-blast) in the frontal cortex. A significant decrease in cerebral blood flow and an attenuated responsiveness of cerebral blood flow to hypercapnia was observed at 2–3 h, 1, 14, and 28 days after blast exposure.

These findings ([109], Chen et al. this Special Issue) cannot be directly extrapolated to rats that received 3 × 74.5 kPa exposures. However, the effects of altering the glycocalyx on cerebral blood flow and the finding that acute behavioral disturbances could be prevented by pretreatment with hyaluronidase suggests a functional role of glycocalyx disruption following blast injury. Further study seems warranted.

## 8. Damage to Smooth Muscle Layers by BINT

High-level blast exposure induces prominent vasospasm in humans [111] and animals [112] along with reduced cerebral perfusion and altered contractile properties of large arteries [113,114,115]. The vascular smooth muscle layer is critical in regulating the contractile properties of blood vessels [70] and also affects the periarterial drainage of soluble substances [71]. The vascular smooth muscle layer is affected by blast exposure acutely after injury in our model and remains so chronically. Figure 2 shows examples of disrupted smooth muscle layers soon after injury.

While we have not studied the effects of 3 × 74.5 kPa blast exposures on smooth muscle-related physiological functions, Abutarboush et. al. [116], in studying rats after a single 130 kPa (18.9 psi) exposure, found functional alterations in the pial microcirculation that evolved acutely from 2 h through 28 days after injury. They observed delayed and prolonged alterations in cerebrovascular reactivity along with changes in the microarchitecture of the vessel wall and perivascular astrocytes. Impaired arteriolar reactivity appeared only >24 h after blast exposure in the form of reduced hypercapnia-induced vasodilation, increased barium-induced constriction, and reversal of the serotonin effect from constriction to dilation. They also saw a reduction in vascular smooth muscle contractile proteins, as well as a delayed reduction in nitric oxide synthase. Collectively, these results suggest that blast exposure results in an evolving pattern of effects on vascular physiology that would be expected to alter long-term vascular reactivity in the NVU.

Why long-term vascular reactivity would be affected is not well understood, although Alford et al. [117,118] have suggested that blast-induced vasospasm may play a role by initiating a phenotypic switch in vascular smooth muscle cells that has long-term consequences. Using high-velocity stretching of engineered arterial lamellae, they found that one hour after a simulated blast, altered intracellular calcium dynamics were seen as well as hypersensitivity to a contractile stimulus with endothelin-1. Interestingly, Abutarboush et al. [116] reported increased endothelin-1B receptors in astrocytes in their in vivo studies.

Later, Alford et al. [117,118] found that one day after a simulated blast, tissues exhibited a prolonged hypercontraction and changes in vascular smooth muscle cells suggestive of a force-dependent phenotypic switch. They argued that blast-induced cerebral vasospasm causes a mechanically induced switch in vascular smooth muscle that potentiates vascular remodeling, leading to further vasospasm and lumen occlusion. Although this phenomenon has yet to be demonstrated in vivo, blast effects on large cerebral blood vessels acutely are well established at higher levels of blast exposure [112]. Whether similar effects occur under low-level blast conditions has yet to be established. Another mechanism would need to be invoked to explain blast effects on capillaries and post-capillary venules where a smooth muscle layer is lacking.

## 9. Low-Level Blast Induces a Gliovascular and Neurovascular Disconnection

In a study aimed at uncovering molecular alterations in the vasculature following BINT, we studied purified brain vascular fractions from blast-exposed animals 6 weeks after exposure using two-dimensional differential gel electrophoresis (2D DIGE) and matrix-assisted laser desorption/ionization (MALDI) mass spectrometry (MS) [35]. We were somewhat surprised to find that the protein showing the most altered expression was glial fibrillary acidic protein (GFAP), which was decreased in blast-exposed animals. Western blotting confirmed that GFAP expression was decreased about threefold in isolated brain vascular fractions from blast-exposed rats (Figure 3A).

GFAP’s presence in isolated vascular fractions, while initially puzzling, seemed to suggest that astrocytic endfeet remain attached to blood vessel walls, with reduced GFAP in blast-exposed samples reflecting a loss of attachments. Indeed, others had previously observed astrocytic endfeet attached to purified mouse brain blood vessels [119,120,121]. Figure 3B shows isolated vascular fractions from control and blast-exposed rats immunostained for GFAP. In controls, GFAP staining was seen in patches, consistent with attached astrocytic endfeet. In blast-exposed rats, perivascular GFAP staining also occurred in patches but was noticeably reduced compared to controls, consistent with astrocytic endfeet being lost. These changes being apparent in Western blots of vascular fractions obtained from whole-brain preparations suggests the widespread nature of the changes.

EM studies showed frequently swollen astrocytic endfeet after blast exposure [35]. Figure 4 shows examples of small arterioles from control and blast-exposed rats at six weeks following blast exposure. Compared to controls, the lumens of blast-exposed vessels appear irregular and thickened. In addition, compared to the tight astrocytic endfeet surrounding the control vessels, perivascular astrocytic endfeet in the blast-exposed vessels are swollen and contain degenerating organelles. Blast-exposed capillaries showed similar changes. Astrocytic changes were more obvious in blood vessels showing the most endothelial cell damage. Although normal-appearing microvessels were apparent in blast-exposed specimens, counts of randomly sampled vessels in the blast-exposed revealed that 18% had clearly swollen astrocytic endfeet [35]. Such changes were not observed in the microvessels of controls.

Equally surprising, in addition to GFAP, of the first nine proteins identified with MS, four more were intermediate filament proteins (IFs), but rather than being glial or endothelial, they were neuronal-associated IFs (α-internexin and the light, medium, and heavy chain neurofilaments) [35]. All were decreased in blast-exposed vessels based on 2D DIGE, observations that were confirmed with Western blotting [35]. In each case, as in the proteomic studies, the levels of neuronal IFs were decreased by about threefold in blast-exposed rats.

Neuronal IFs in isolated vascular fractions suggested that neuronal fibers remain tightly associated with purified brain vascular preparations. Others have also observed neuronal IFs in purified preparations of mouse brain vasculature [119,120]. Immunostaining of vascular fractions for heavy neurofilament (NFH) showed focal staining sometimes with fine processes apparent, consistent with neuronal attachments to blood vessels. Their decrease following blast injury suggests that, like gliovascular, neurovascular attachments are being disrupted as a consequence of blast injury.

Interestingly, vascular fractions from animals taken at 8 months post-blast exposure showed normal levels of GFAP with Western blotting suggesting perhaps some attempt at re-establishing astrocytic connections to blood vessels [35]. Yet, a chronic vascular pathology was still visible with EM. A phenomenon frequently observed, however, was GFAP-containing processes within the vascular lumen (Figure 2). We suspect that intraluminal GFAP appears when astrocytic processes regrow towards a damaged vessel but are not able to establish normal connections. Neuronal IFs also become normalized in vascular fractions isolated from rats at 8 months after blast exposure. Collectively, these observations suggest that normalization of GFAP and neuronal IF levels in isolated vascular fractions, while signaling a regrowth of astrocytic and neuronal processes, does not translate into a restoration of a normal vasculature structure [35].

Loss of GFAP in isolated vascular fractions seems to be a general phenomenon in rodents and is not limited to blast-exposed rats. Similar to blast-exposed rats, we have found that GFAP was decreased by about twofold in isolated vascular fractions from blast-exposed mice harvested at four weeks after a repetitive blast exposure protocol [122] (unpublished observations). Thus, blast exposure induces a gliovascular disconnection in rats and mice.

Lost gliovascular connections also appear to occur acutely after injury as Hubbard et al. [92] have shown in male and female rats exposed to single 75.8 kPa (11 psi) peak overpressure exposures. They were further able to show that astrocytic endfoot coverage around blood vessels was decreased in males. Interestingly, by 7 days, GFAP levels had recovered, suggesting that damage following single exposures may recover while multiple repetitive exposures, delivered over-short intervals, as in our model, lead to damage that does not recover [92].

Several studies from other labs have described ultrastructural changes in astrocytic endfeet following BINT [82,123,124]. In a rat model of blast injury that used live explosives (110 kg TNT equivalent), Kaur et al. [123] observed astrocytes in the cerebral and cerebellar cortex with enlarged and swollen endfeet. Lu et al. [124] described enlarged and “watery” astrocytic endfeet in a non-human primate model also utilizing live explosives reported as 80 kPa exposures. These changes, however, occurred in the context of a broader neuronal, glial, and microglial pathology that was absent in our model. Another study described swollen astrocytic endfeet in a mouse model of 77 kPa blast injury, although pathology in this model, which did not restrain the head, occurred in the setting of a broader neuropathology with neuronal tau pathology, axonal changes, and widespread astrogliosis [82].

Supporting the relevance of the animal studies to humans, serum GFAP levels have been found elevated acutely in military personnel after heavy weapons training [125]. Levels remained elevated for at least 3 months in this study [125], although in another study serum, GFAP levels were transiently reduced in military personnel on days 6 and 7 of a 2-week training protocol that involved exposure to moderate blasts [9]. In this second study, GFAP levels negatively correlated with cumulative blasts experienced during training and with duration of military service [9]. Shively et al. [126] have also described a distinctive pattern of interface astroglial scarring at the grey/white matter junctions that may be unique to BINT. The reasons for the variations in time course and the relationship of any of these changes to vascular injury is unclear.

## 10. Functional Consequences of Lost Gliovascular and Neurovascular Connections on Blood–Brain Barrier Function and Neurovascular Coupling

Perivascular astrocytes play many roles which, by their loss or disruption, could influence the effects of BINT. One of the most important functions of astrocytes is the formation and maintenance of the BBB, a role that may significantly affect the uptake of nutrients, including glucose, hormones, lipids, and amino acids. Astrocytes support BBB integrity by promoting the maintenance of tight junctions between the endothelial cells of blood vessels [60,61,62].

Studies in the 1980s found that in coculture experiments, astrocytes induced BBB properties in brain endothelial cells [60]. Furthermore, when glial progenitors from the CNS were transplanted into peripheral tissues, CNS progenitors induced BBB properties in peripheral vessels that included tight junctions between peripheral endothelial cells which prevented tracer leakage as in the CNS [127]. It would be later shown that an astrocyte-conditioned medium could induce BBB features in endothelial cell cultures, arguing that secreted factors drive BBB formation and maintenance [60]. Astrocytes promote the expression of a number of junctional proteins, enzymatic pathways, and specific transporters, including the glucose transporter-1 (GLUT-1) [128,129,130,131].

Recent studies have attempted to directly assess the role of perivascular astrocytes using laser ablation [60]. Simultaneously ablating adjacent astrocytes caused discontinuous endothelial tight junctions and irregular basement membranes [132]. Astrocyte ablation in the spinal cord reduced expression of the tight-junction protein occludin [133]. However, in these studies, substantial functional BBB damage was not observed based on erythrocyte or fibrinogen extravasation [133,134].

Live in vivo imaging during and after the ablation of single astrocytes or endfeet has shown that these structures have considerable plasticity, with neighboring astrocytes quickly recovering vessel surfaces [60,135,136]. Furthermore, without astrocytic endfoot coverage, vessels showed no leakage of plasma marker dyes such as Evans Blue or dextran-conjugated fluorescein, while when the vascular wall was ablated, Evans Blue extravasation was seen. These studies question how much blast-induced disruption of astrocytic endfeet might be expected to affect BBB integrity, although failure to affect permeability to dyes such as Evans Blue or fluorescein-conjugated dextrans does not preclude that specific transport systems that move smaller solutes including ions, lipids, glucose, and small proteins may be affected.

Astrocytes sense neuronal activity and regulate vascular tone in the context of neurovascular coupling [52]. Glutamate released from neurons acts on astrocytic metabotropic glutamate receptors to evoke the Ca^2+^-dependent release of vasoactive metabolites from astrocytic endfeet. Under normoxic conditions, astrocytic Ca^2+^ signaling results in vasodilation, whereas under hyperoxia, vasoconstriction is favored [52].

Cerebral pial vessels are innervated by perivascular nerves from both autonomic and sensory ganglia of sympathetic, parasympathetic, and trigeminal origins [137]. Sympathetic terminals have significant vasoconstrictor effects in large cerebral arteries. Under conditions of elevated hydrostatic pressure, sympathetic vasoconstriction protects against increases in venous pressure, BBB disruption, and edema formation. Stimulation of parasympathetic nerves has potent vasodilator effects on cerebral arteries, resulting in increased blood flow. Thus, BINT-induced alterations in neurovascular connections could disrupt cerebral autoregulation at multiple levels.

## 11. BINT and Glymphatic Function

The brain produces substances that seem to be best considered as waste products. That material is primarily cleared along cerebrovascular routes, either by direct transport across the BBB into the vascular lumen or via flow along perivascular channels with drainage into the lymphatics [138]. The glymphatic system is a waste-clearance system of perivascular channels formed by astrocytes [65,139]. Cerebrospinal fluid pumped into the brain along penetrating arterioles mixes with interstitial fluid and waste products and is cleared out of the brain along venules [65]. In addition to waste products, the glymphatic system may also help transport non-waste substances including neurotransmitters, glucose, lipids, and amino acids. The glymphatic system has particularly gained attention for its role in transporting potentially neurotoxic waste products involved in neurodegenerative diseases, such as amyloid beta (Aβ) and tau proteins [140,141,142].

An alternate intramural periarterial drainage system is also believed to exist that moves interstitial fluid containing waste products toward the pial surface along the basement membranes of arterioles in the opposite direction of blood flow [71]. Astrocyte endfeet are proposed to drive intramural periarterial drainage by regulating functional hyperemia and cerebrovascular tone [143]. Thus, astrocytes could regulate perivascular flow under either intramural periarterial drainage or glymphatic.

TBI, in general, is a risk factor for the later development of neurodegenerative diseases that may have varied underlying pathologies [144,145,146]. Proteins associated with neurodegenerative diseases, including Aβ, and tau accumulate in brain following TBI [147]. Aβ deposition is a hallmark of Alzheimer’s disease (AD) and epidemiological studies support an association of severe TBI with a later risk for developing AD [146,148]. Acutely, Aβ levels rise rapidly after TBI, with increased levels of soluble Aβ and diffuse cortical deposits present in humans as early as two hours after a severe injury [146,148,149,150].

Amyloid precursor protein (APP) and Aβ elevations occur acutely in brain in many experimental animal models that mimic the type of contusional and rotation/acceleration injuries associated, for example, with motor vehicle accidents or sports injuries [151,152,153,154,155,156,157,158,159,160,161,162]. In these models, there is increased expression of APP, along with BACE-1 (β-site APP-cleaving enzyme-1), the principal α-secretase and the γ-secretase complex responsible for the C-terminal cleavages that together are responsible for generating the 39–43 amino acid peptide Aβ [152,163,164,165,166]. Upregulation of amyloidogenic APP processing, which favors Aβ production over other non-amyloidogenic APP processing [167], has been suggested as an explanation for the epidemiological associations between TBI and AD [146,148].

Widespread disruption of gliovascular connections following blast exposure seems to predict that glymphatic flow would be widely disrupted and reduce Aβ clearance. While this has not been studied in our chronic rat model of repetitive blast exposure, in a study in both rat and mouse models of acute blast exposure, rather than increasing, rodent brain Aβ42 levels decreased in the first week following blast exposure [32]. Interestingly, the effect on Aβ42 was most prominent in rats exposed to lower blast overpressures (36.6 kPa and 74.5 kPa), while there were no effects on Aβ42 at the 116.7-kPa exposure level [32].

In a later study [168], we subjected APP/presenilin 1 transgenic (Tg) mice (APP/PS1 Tg), a transgenic mouse model of AD, to an extended sequence of repetitive low-level blast exposures (34.5 kPa, 5 psi) designed to mimic human subclinical blast exposures. Exposures were administered three times per week over 8 weeks beginning at either 20 or 36 weeks of age, which represent times before and after this line of APP/PS1 Tg mice develop significant plaque burdens [169].

Curiously, repetitive exposures reduced anxiety as well as improved cognition and social interactions in APP/PS1 Tg, returning many behavioral parameters in APP/PS1 Tg mice to the levels of non-transgenic, wild-type mice [168]. Effects were most apparent in APP/PS1 Tg mice that received blast exposures beginning at 20 weeks of age. Results were less robust in mice when blast exposure began at 36 weeks of age, likely reflecting the greater difficulty of reversing behavioral deficits in mice with more extensive amyloid burden [169]. While amyloid plaque loads were unchanged, soluble, insoluble, and oligomeric Aβ42 levels were reduced in the brains of mice exposed to repetitive low-level blast exposure [168].

More recently, Abutarboush et al. [170] explored the mechanistic basis for these effects in rats by examining the effects of a single 37 kPa (5.4 psi) blast exposure on brain Aβ levels, production, and clearance mechanisms in acute (24 h) and delayed (28 days) phases after blast exposure. Aβ, including the neurotoxic detergent soluble Aβ42 form, was reduced at 24 h but not 28 days after blast exposure. Aβ42 reduction was not associated with changes in levels of Aβ oligomers, levels of APP, or in the β- and ϒ-secretases BACE-1 and presenilin-1, which are involved in the amyloidogenic cleavage of APP. Levels of the α-secretase ADAM17 (also known as tumor-necrosis factor α [TNFα]-converting enzyme), which promotes the non-amyloidogenic processing of APP, were decreased. Levels of BACE-1, and the γ-secretase component, presenilin-1, were also unchanged in our prior study of acute blast exposure in rats [32].

In Abutarboush’s study [170], contrasting with the lack of effects on the enzymes involved in the processing of APP into Aβ, at 24 h after blast exposure, brain levels of the endothelial transporter, low-density related protein 1 (LRP1) increased, and there was enhanced aquaporin-4 (AQP4) localization on perivascular astrocytic endfeet. These findings seem to best explain lowered brain Aβ levels after blast exposure as due to enhanced transvascular endothelial clearance of Aβ by LRP1-mediated transcytosis and altered AQP4-aided glymphatic clearance, suggesting that a low-intensity blast alters transvascular and perivascular Aβ clearance. Future studies will be needed to understand whether the same mechanisms apply to APP/PS1 Tg mice or rats after chronic repetitive exposures.

Few other studies have examined Aβ levels after blast exposure. Konan et al. [88], using a mouse low-level blast exposure protocol, found Aβ40 and 42 elevated at 30 days but not at 7 days after exposure. Harper et al. [171] examined the chronic effects of blast TBI on retinal ganglion cells, optic nerve, and brain amyloid load in APP/PS1 Tg mice and non-transgenic littermates exposed to a single 20 psi blast at 2 to 3 months of age. Two months later at ca. 4.5 months, the APP/PS1 Tg mice exposed to the blast had poorer retinal function, thinner retinal ganglion cell layers, and more optic nerve damage than sham-exposed APP/PS1 Tg mice or blast-exposed non-transgenic mice [171]. No Aβ plaque deposits were found in the retinas of APP/PS1 mice, although increased APP/Aβ-immunostaining was found in blast- compared to sham-exposed transgenic mice, particularly in perivascular regions. They found a statistically non-significant trend towards greater cerebral cortical Aβ plaque loads in the blast-exposed mice compared to the sham-exposed transgenic mice [171].

Studies in U.S. military personnel have largely mirrored results in animal studies by showing decreased Aβ following blast exposure. Edwards et al. [172] found that during a 10-day training exercise that involved repeated blast exposure, Aβ42 was lowered in blood at 24 h following blast exposure [172]. Transient reductions in APP and alterations of the APP signaling network in blood were also observed during training exercises that involved a moderate blast exposure [173]. These studies suggest that as in experimental animals, altered APP processing is an effect of acute blast exposure in humans, although another study found elevated serum Aβ42 in military personnel who experienced repeated blast exposures from firing 0.50-caliber rifles in training sessions conducted over multiple days [174]. In addition, chronically elevated serum levels of Aβ40 and Aβ42 have been found in healthy, male, active-duty military and law enforcement personnel exposed to regular military occupational blasts at U.S. Department of Defense and civilian law enforcement training sites [175]. Thus, effects in humans may vary with type and intensity of exposure as well as the temporal relationship to testing. Further studies will be needed to understand these relationships.

## 12. Perivascular Tau following BINT

Chronic traumatic encephalopathy (CTE) is a progressive cognitive/behavioral syndrome that most often follows repetitive mTBI [144]. A few reports have appeared of pathologically confirmed CTE in military veterans with a history of blast exposure [82,176,177]. Increased plasma tau has been seen in recently deployed veterans with a history of TBI, with correlations between levels and symptom severity [178]. Elevations of neuron-derived tau in extracellular vesicles have also been observed in serum from experienced breachers where it correlated with neurobehavioral symptoms [179]. Three studies have observed increased retention of the PET tau ligand [^18^F]AV1451 (flortaucipir) either in recent military veterans or Vietnam-era veterans who suffered from chronic cognitive and neurobehavioral syndromes after blast injury [33,180,181]. The prevalence of CTE pathology in veterans with chronic cognitive/behavioral syndromes following blast exposure remains to be determined, although a large post-mortem study found, in general, little association between blast-related TBI and CTE pathology post-mortem [182].

Our model of repetitive low-level blast exposure develops pathological accumulations of hyperphosphorylated tau (p-tau) in neuronal perikarya and perivascular astroglial processes in multiple brain areas [33]. These deposits are present at least as early as six weeks after blast exposure and remain present at one year [33]. Examples of perivascular deposits of p-tau are shown in Figure 5. The accumulations using Western blotting showed a regional pattern in that p-tau was increased in the anterior neocortex and hippocampus but not in the amygdala [33].

Tau’s phosphorylation pattern is complex, with over 40 known sites [183,184] and changes in tau phosphorylation following blast exposure were specific to certain p-tau sites [33]. Following blast exposure, we found increased phosphorylation at p-Thr181 [33], which is hyperphosphorylated in human paired helical filaments (PHF) tau [185]. Increased phosphorylation was also seen with the antibody CP13, also raised against PHF tau from human AD brain and recognizing p-Ser202 [33]. No changes were found at three other phosphorylation sites (Thr231, Ser396, and Ser404) that either inhibit microtubule stabilization (Thr231) or promote tau’s self-association properties and oligomerization (Ser396, Ser404) [183,184]. The presence of two phosphor-sites found in human PHF tau (p-Thr181, p-Ser202) following blast exposure suggests a pathological character to the p-tau deposition. Interpreting how the pattern of tau phosphorylation after blast exposure affects tau aggregation and microtubule function more broadly is difficult from the available data.

A puzzling aspect of our rat model is the laterality of blast effects on tau [33,186]. If a brain area was affected, the right side was always involved with the corresponding left side only affected if the right was as well. Particularly striking were the asymmetrical effects on the right and left hippocampus. p-Tau levels further correlated with blast-induced behavioral traits [186]. Elevated p-tau Thr181 in anterior neocortical regions and right hippocampus correlated with anxiety as well as fear learning and novel object localization. By contrast, there were no correlations with levels in the amygdala or posterior neocortical regions and levels did not correlate with hyperarousal. Results were specific to Thr181 in that no correlations were observed for three other sites (Thr231, Ser396, and Ser404) and no consistent correlations were linked with total tau.

Since the blast exposure in our model is delivered as a straight frontal exposure, no systematic variation in the rat’s placement within the blast tube should cause the right hemisphere to be differentially impacted. We can only speculate that this pattern reflects some laterality of hemispheric function in rats, which causes the right side to respond differently to the shock wave or is a feature of the evolution of the injury. Although not as well appreciated as in humans, rats and mice have been known to exhibit paw preference and hemispheric laterality for complex behavioral functions since the 1970s [187]. Hemispheric dominance, in particular, affects spatial memory in rodents [188,189]. Dopaminergic systems, which are disrupted by blast exposure [190] also exhibit lateralization patterns [191,192,193].

Biological models of PTSD postulate that alterations in frontal and limbic structures, including the prefrontal cortex, amygdala, and hippocampus, are involved in generating exaggerated fear responses in humans [194,195,196], which are also seen in blast-exposed rats [26]. Functional neuroimaging in humans is consistent with such models, suggesting that in PTSD, heightened amygdala activity is associated with decreased hippocampal and orbitofrontal activity [194].

p-Tau in our rat model was increased in the anterior cortex and hippocampus, including prelimbic cortex, but not in the amygdala or posterior cortex [33,186]. These correlations are significant in suggesting that p-tau accumulation in anterior neocortical regions and hippocampus may lead to disinhibited amygdala function without p-tau elevation in the amygdala itself. If an evolving tauopathy reduces prefrontal and hippocampal inhibition of the amygdala then, as in biological models of PTSD, this could lead to enhanced fear responses in blast-exposed rats [26] suggestive of amygdala hyperactivity.

Interestingly, in our rat model at six weeks after blast exposure, p-tau was increased in the right anterior neocortex but not in the left [33]. By contrast, at 10 months, it was increased on both sides. This evolution has a relationship to the developing PTSD-related behavioral phenotype, which is absent before 6 weeks but present after 12 weeks [27]. One can speculate that this “spread” of p-tau to both hemispheres marks the critical point at which frontal inhibition of the amygdala fails and onset of the fear- related phenotype begins.

Correlations with fear-related behavior and p-tau in the right hippocampus also suggest a role for hippocampal tau in progression of the behavioral phenotype [186]. Whether in this context p-tau is serving as a marker of neurodegeneration or an actual spread of tau is occurring, as is believed to happen in many neurodegenerative diseases, [197,198,199] is unclear, but seems worthy of further study. Some in vitro evidence supports the notion that astrocytes may be a mediator of cell-to-cell spread of pathological tau [199]. If applicable to blast pathophysiology, this mechanism could have relevance as to how an initial vascular injury with perivascular astrocyte injury leads to a later behavioral phenotype that must have a neuronal basis.

Tau changes might be more broadly involved in the behavioral phenotype of blast- exposed rats in that stress affects phosphorylation at Thr231, Ser262, and Ser396/404 [200]. Tau also seems essential for mediating stress responses [200,201]. This was clearly demonstrated by Lopes et al. [201], who subjected tau-null mutant mice to chronic, unpredictable stress. Atrophy of apical dendrites and spine loss in prefrontal cortical neurons as well as impairments in working memory induced by chronic stress in wild-type mice was absent in tau-null mutant animals [200,201].

p-Tau in perivascular astrocytic processes following blast exposure is of interest in comparing blast pathology to CTE. In CTE, p-tau aggregates occur in neurons and astrocytes in perivascular locations, especially within the depths of the cortical sulci [202,203]. However, recent consensus statements have emphasized the pathognomonic perivascular lesion in CTE as p-tau in perivascular neurons [202,203,204]. p-Tau in astrocytes, while present, is considered a less consistent feature of CTE. Perivascular neuronal p-tau is also apparent in blast-exposed rats [33].

Altered tau processing in the form of increased p-tau has frequently been observed in experimental animal models of blast injury [82,87,88,205,206,207,208,209,210,211,212,213,214,215,216,217,218,219,220,221,222,223,224,225,226]. Two studies reported tau oligomers following blast exposure [218,220]. In mice, p-tau accumulations have been seen in retinal neurons and retinal glia following blast injury [223]. Tau deposits have also been observed in the superficial layers of the mouse cerebral cortex [82].

Meabon et al. [217] described perivascular accumulations of tau in a mouse model of blast injury. These accumulations appeared within hours of injury. Once present in our model, perivascular tau accumulations remain late in the course of the disease [33]. Thus, perivascular tau accumulation seems to be an early and persistent feature of the injury.

Astrocytes influence a diverse variety of animal behaviors through their interactions with neurotransmitters released in perisynaptic regions and their role in regulating calcium currents in the brain [227]. How much of a role tau in perivascular astrocytes plays after blast injury remains to be determined. Brain responses to blast exposure can be conceptualized as a type of stress response, making it of interest to determine how blast responses would be altered in tau-null mutant mice or rats.

## 13. BINT Damage to Pericytes and the Extracellular Matrix

Pericytes and their normal association with other components of the NVU are disrupted following blast injury in the context of the general disruption of the NVU seen following BINT [37,38]. Damage to pericytes in the walls of capillaries and post-capillary venules could contribute to any number of altered functions that have been attributed to pericytes including BBB maintenance, regulation of immune cell entry into the CNS, and neuroinflammatory states [66]. Neuronal activity may also regulate capillary blood flow through Ca^2+^-dependent astrocytic mechanisms that actively dilate and constrict pericytes in response to vasoactive agents [228,229]. Pericyte degeneration leads to neurovascular uncoupling and limits oxygen supply to the brain [230]. What specific role pericyte disruption plays in BINT in the context of a more generally disrupted NVU remains to be determined.

The vascular extracellular matrix exists as a complex mixture of proteins, proteoglycans, and other structural proteins that include those in the basement membranes but extend well beyond the immediate perivascular region [231,232]. As a reservoir for growth factors and adhesion proteins that regulate BBB permeability [57,59], disruption of the vascular extracellular matrix could affect a variety of functions within the NVU following blast injury.

In studies investigating whether blast exposure induced changes in the vascular extracellular matrix in our rat model, we performed immunohistochemical staining for vascular extracellular matrix components including collagen IV and laminin [34,37]. We and others have found that efficient immunostaining of collagen IV and laminin in mature adult rodent brain requires a proteolytic epitope unmasking step in perfusion-fixed tissues [233]. In wild-type adult brains, widespread immunostaining for collagen IV and laminin can only be reliably detected following pepsin pretreatment [233]. However, under certain pathological conditions that include vascular degeneration, immunodetection of collagen IV and laminin occurs without antigen retrieval [234].

In blast-exposed animals examined at six weeks after exposure, immunostaining of vascular elements with laminin and collagen IV was observed without pepsin pretreatment [34,37] with areas of enhanced staining that extended broadly across the cortical regions. Pathological staining for collagen IV (Figure 6) and laminin was present at 10 months and longer after blast exposure, suggesting that an altered vascular extracellular matrix is present at subacute stages and a chronic feature of injury [34,37].

Vascular immunostaining with collagen IV or laminin in the brains of blast-exposed rats without pepsin digestion argues for a reorganization of the vascular extracellular matrix resulting in increased accessibility of these antigens. Increases in matrix metalloproteinases 2 and 9 (MMP-2 and MMP-9) have been observed in isolated vascular fractions at six weeks after blast exposure [37]. Elevation of MMP-2 and MMP-9 suggests that elevation of endogenous degrading proteases may be the basis for these alterations [37]. Others have observed changes in matrix metalloproteinases acutely following blast exposure in animals [235,236,237]. In humans, levels of serum MMP-9 have been found elevated acutely and then for at least 3 months in military personnel during heavy weapons training [125].

Interestingly, the staining of vessels with antigens such as collagen IV without pepsin treatment is typical of immature vessels [233] and seen in vessels in injured areas of the adult brain following mechanical trauma [238], as well as in some models of neurodegeneration [234]. The physiological basis for these differing patterns of staining is incompletely understood, but seems to correlate with the tightness of gliovascular junctions (reviewed in [233]). Indeed, one major structural difference between immature and mature microvessels in the brain is the large perivascular space present in immature microvessels that separates the outer glial basal lamina and the inner vascular one [238,239].

The tightness of gliovascular junctions correlates with the intensity of laminin immunostaining following mechanical injury [238]. Thus, by analogy, altered collagen IV and laminin staining following blast exposure may reflect the loss of normal tightness of gliovascular junctions after blast exposure [35] and could be interpreted as evidence of increased vascular remodeling. Others have observed decreases in the vascular junctional proteins occludin, claudin-5, and ZO-1 acutely following blast exposure [92,122,240,241,242,243,244]. In humans, levels of occludin and claudin-5 are elevated acutely and then for at least 3 months in military personnel during heavy weapons training [125], changes that suggest a loss of vascular tight junctions in human low-level BINT.

Extracellular matrix proteins play many functional roles in the brain, but among their potentially most far-reaching is their role in perineuronal nets (PNNs) [245,246]. PNNs are specialized structures responsible for synaptic stabilization in the adult brain. PNN development plays a critical role in the closure of critical developmental periods associated with high plasticity [245,246]. The digestion of PNNs in adult tissue can restore critical period-like synaptic plasticity [245,246]. Key molecular components of PNNs are chondroitin sulfate proteoglycans (CSPGs), hyaluronan, hyaluronan- and proteoglycan-link proteins (HAPLNs), and the glycoprotein tenascin-R, some of which, as discussed above, are also found in the vascular glycocalyx, which is affected acutely by BINT [109]. PNNs form reticular meshes that surround the perikarya and proximal dendrites of subsets of neurons. PNNs have been most studied in relation to parvalbumin-expressing inhibitory neurons. However, projection neurons, including some hippocampal pyramidal cells, possess PNNs [247,248].

Evidence suggesting that PNNs regulate synaptic plasticity in the adult has come from experiments showing that ocular dominance plasticity could be restored after treatment with chondroitinase ABC (ChABC), an enzyme that degrades CSPGs [249]. Other studies have suggested a role for PNNs in memory retention [245] and PNNs have been implicated in regulating fear-related memory and anxiety [250,251,252,253,254,255]. While initial attention focused on PNNs in the amygdala [251], PNN disruption in the hippocampus affects fear learning in rats [253]. Disruption of PNNs in the medial prefrontal cortex impairs some forms of fear memory as well [253] and intrahippocampal injection of hyaluronidase has been reported to impair contextual fear memory in mice [254]. Given the alterations seen in the extracellular matrix, PNNs seem to warrant exploration in the context of BINT.

Collectively, these observations suggest that blast exposure induces degradation and remodeling of the extracellular matrix. Why vascular remodeling would continue many months after blast exposure, as suggested by our findings in rats, is unknown. Further examination of the molecular changes occurring, particularly in the chronic phase, will be needed to understand the pathophysiological basis of this injury. Functional studies will also be needed to determine whether BBB function is disrupted as well.

## 14. Vascular-Associated Neuroinflammation following BINT

Microglia have macrophage-like properties and represent 10% of cells in the brain [256]. While not formally part of the NVU, microglia are found near and sometimes contacting blood vessels, as well as synapses, dendritic spines, and processes of perisynaptic and perivascular astrocytes. Microglia support brain health by removing tissue debris and pathogens, as well as remodeling synapses [257].

After injury, microglia are activated, becoming proliferative and undergoing morphological and physiological changes that are associated with increased phagocytic activity and proinflammatory cytokine secretion [256]. This results in the propagation of inflammatory responses that may, in some cases, contribute to brain pathology, although microglia also play roles in maintaining vascular integrity and vascular repair [258].

Microglial morphologies can be correlated with different activation states [259,260,261,262,263,264]. Functionally, microglial activation can be divided into separate states [265]. A pro-inflammatory M1 state is associated with neurotoxicity and extracellular matrix damage. By contrast, the M2 state is associated with phagocytic, anti-inflammatory, and neuroprotective effects that promote processes such as wound healing [265].

We have examined inflammation in our model of low-level blast exposure in several previous studies [36,37,38]. In a study that focused on changes 6 weeks after blast exposure, we found no clear morphological evidence for inflammation in the brain [36]. In this same study, an analysis of 27 cytokines and growth factors in plasma and four brain regions, including the right and left anterior and posterior neocortex, hippocampus, and amygdala, was conducted at 6 and 40 weeks after blast exposure. While a handful of cytokines were decreased in the plasma at 6 weeks (fractalkine, interleukin-1β, lipopolysaccharide-inducible CXC chemokine, macrophage inflammatory protein 1α, and vascular endothelial growth factor), none of the changes in plasma were reflected in brain. No differences in plasma cytokine levels were detected between blast-exposed and control rats at 40 weeks after blast exposure. In the brain, isolated changes were seen in levels of selected cytokines at 6 weeks following blast exposure, but none of these changes were found in both hemispheres or at 40 weeks after blast exposure [36]. The conclusion of this study was that inflammation is not a prominent feature in our model at 6 weeks after blast exposure.

In a later study, we focused on inflammatory changes at one year and longer after blast exposure [38]. This study found patchy neuroinflammatory changes in areas where there were vascular alterations. At 13 months after blast exposure, small perivascular patches of activated microglia expressing high levels of major histocompatibility type II antigens (MHCII) were often observed (Figure 7), a feature characteristic of M1 microglia (the proinflammatory type associated with tissue damage). Morphologically, the microglia were predominantly the more pro-inflammatory types 3 and 4. Some perivascular microglia were visibly undergoing apoptosis and evidence for perivascular microglial phagocytosis of astrocytes was found in the form of intracellular GFAP-immunostained material within perivascular microglia. A variety of EM changes were seen in perivascular microglia, including dilated endoplasmic reticulum, mitochondrial alterations, and cholesterol crystals in lysosomal vacuoles, all suggestive of phagocytic activity [38].

The results of our early [36] and late studies [38] suggest that perivascular neuroinflammation develops after 6 weeks following blast exposure. However, once established, it likely persists into the chronic phase. This time frame correlates roughly with the appearance of the blast-related behavioral phenotype, which is absent before 8 weeks and present 12 weeks and longer after blast exposure [27]. The patchiness of the morphologically visible neuroinflammatory response, however, limits direct correlations [38]. While in individual animals, focal patches of inflammation seemed to bleed into more widespread strips of inflammation that were associated with neuronal and synaptic loss, this was not widely found in the brain [38].

A more general suggestion of a blast-associated inflammatory response was found in a transcriptomic study that examined blast-induced alterations in the anterior neocortex, hippocampus, amygdala, and cerebellum across the time frame over which the PTSD-related behavioral phenotype develops [40]. This study found a group of differentially expressed genes (DEGs) regulated by tumor necrosis factor α (TNFα) whose expression was altered in a direction consistent with TNFα activation in the amygdala and anterior cortex between 6 weeks and 12 months after blast exposure [40].

A group of 6 DEGs (*Hapln1*, *Grm2*, *P2ry12*, *Ccr5*, *Pbld1*, and *Cdh23*) were identified in blast-exposed animals that were common to the anterior neocortex, amygdala, and hippocampus. Of these, the purinergic receptor *P2ry12* is of particular interest, since it is highly expressed in quiescent non-inflammatory M2 microglia, but its expression is downregulated in activated M1 inflammatory microglia associated with neuroinflammation [266]. *P2ry12* has been identified as one of a set of genes highly enriched in microglia that can be used for phenotyping microglia [267]. The widespread downregulation of *P2ry12* in multiple brain regions from studies conducted on whole-tissue extracts suggests a broader inflammatory signal after blast exposure that may not be associated with morphological evidence of inflammation.

Inflammation has long been suspected of playing a role in BINT [31]. Inflammatory changes have been studied mostly following acute blast exposure [31]. In animal models, blast exposure causes increases in many markers of inflammation within blood and plasma [31]. Microglial activation has been reported in many studies [31,42,82,85,123,206,268,269,270,271,272,273]. A recent study by Stone et al. [274] demonstrated increased brain inflammation in special forces personnel using positron emission tomography (PET) to detect uptake of the translocator protein (TSPO) ligand DPA-714, which recognizes activated microglia. Increases in brain inflammation were associated with decreased cortical thickness and volume changes and were positively associated with lifetime repetitive blast exposure.

In other animal studies, elevations of multiple pro-inflammatory cytokines and chemokines including TNFα have been observed in multiple brain regions at 5 days after a single 70 kPa low-level blast exposure in rats [84]. TNFα has been elevated in many studies following blast injury in rodents [84,94,275,276,277,278,279], although in some of them, elevations were transient returning to baseline within 48 h of injury [275,276]. Simard et al. [102] observed perivascular induction of TNFα in their studies of selective vascular exposure.

Interestingly, TNFα in the brain has biological activities independent of its role in inflammation, being implicated in PTSD-related hyperarousal states and regulating fear learning [280,281,282,283,284]. For example, increased TNFα blocks the retrieval and reconsolidation of contextual fear memories [280], while intracerebroventricular injection of TNFα impairs fear extinction [282]. TNFα signaling in the hippocampus has also been implicated as mediating enhanced fear learning during drug withdrawal [281] and maternal TNFα production is suggested to program innate fear responses in offspring [284]. The cellular basis for these effects is unclear, although microglial production of TNFα has been implicated in sustaining fear memories [283]. Thus, a variety of evidence supports a potential role for TNFα signaling in the neurobiological basis of blast effects on behavior through inflammatory or possibly non-inflammatory mechanisms.

Human studies in experienced breachers exposed to high numbers of career blast exposures also show the dysregulation of genes associated with chronic inflammation and immune response in blood [285] and have identified elevated TNFα levels in neuronal-derived extracellular vesicles isolated from serum [286]. TNFα is a therapeutic target in many human inflammatory conditions and has revolutionized the treatment of some [287,288]. Given its role as a key mediator of brain damage in many conditions, including brain trauma [289,290], exploration of its role in BINT is warranted.

## 15. Linking Neurovascular and Chronic Neurobehavioral Effects That Follow BINT

At the levels of blast exposure we are studying, the initial injury appears largely vascular. The question then becomes, could an initial vascular injury contribute to a delayed and chronic neurobehavioral phenotype that must have a neuronal basis? It is possible that some neuronal injury hidden from sight initiates a strictly neuronal sequence of events leading to the behavioral phenotype. All we can say at this point is that if there is, it is not yet inapparent, while an early vascular pathology is readily apparent.

In this review, we have highlighted the various injuries to the NVU that might play a role in a vascular hypothesis suggesting that an initial vascular injury could initiate a sequence of pathophysiological events leading to a later behavioral disturbance. These changes are summarized and presented in relationship to the late behavioral phenotype’s appearance in Table 2 and Figure 8. Broadly, this model suggests that an early and selective vascular pathology occurs with direct endothelial injury. Associated with endothelial injury there is damage to, and loss of, perivascular astrocytes, altered vascular extracellular matrix proteins, and damage to the smooth muscle layer and other mural cells. Perivascular tau accumulation is also an early feature of the injury, and each of these factors potentially has different contributions to the resulting NVU pathology (Table 2).

Effects at the level of the glycocalyx, endothelial cells, perivascular astrocytes, and nerves, as well as the pericyte and smooth muscle layer, could all contribute to a disturbed NVU affecting a variety of vascular-related functions, including BBB permeability, neurovascular coupling, and glymphatic flow. A perivascular tauopathy might have consequences directly on vascular function or indirectly, if it served as a seed for subsequent neuronal spread of tau.

Of the primary elements within the NVU that are disrupted by BINT, perivascular astrocytes, given their diverse effects on the NVU, emerge as particularly good candidates for playing an important role. Their loss or disruption could influence BBB properties, nutrient uptake, neurovascular coupling, and immune cell entry into the CNS. Astrocytes are also attractive because of their broader role in regulating neuronal activity as part of the tripartite synapse [227]. Astrocytes are activated by and clear synaptically released neurotransmitters, particularly glutamate, and intracellular Ca^2+^ signaling in astrocytes regulates neuronal activity at distant sites [227]. With their broad anatomic reach and well-known effects on behavior, astrocytes are attractive as mediators of an initial vascular injury to distant sites.

Among the secondary factors that a vascular injury might promote, neuroinflammation is attractive as a linking mechanism between vascular injury and a chronic behavioral phenotype. Substantial evidence over many years has established chronic, low-grade inflammation as a feature of many mental health disorders including depression and PTSD as well as a pathophysiological driver of symptoms [291,292,293,294,295,296].

Haroon et al. [297] suggested an interesting model proposing that in mood disorders, initial perivascular inflammation is a seed that leads to later glutamatergic dysfunction and mood disorders. In this model, perivascular inflammation is postulated to convert the synaptic environment from a homeostatic one into a toxic inflammatory one, leading to a hyperglutamatergic and excitotoxic environment [297]. While in mood disorders there is no established vascular lesion that might serve as a seed to trigger such a cascade, this model seems attractive for low-level blast injury where evidence for an initial vascular injury is well established.

Supporting an early, potentially hyperglutamatergic, state following blast injury, a number of studies have shown increased evoked synaptic NMDA receptor activity in hippocampal slices taken from blast-exposed rats in vivo [298,299] or in vitro [300], with one study [299] showing increases still present six weeks after blast exposure. Collectively, a vascular injury followed by perivascular astrocyte injury and neuroinflammation might explain how an injury that starts in the NVU spreads to involve neuronal circuits and produces a behavioral phenotype that must have a neuronal basis.

While the neurobiological basis for the behavioral traits in our model are not fully understood, our most recent work suggests that the late behavioral traits seem to be caused by abnormalities in the group II metabotropic receptor mGluR2 [301]. mGluR2 is elevated after blast exposure in a time course that correlates with the appearance of the behavioral phenotype [27,301]. A causative role for mGluR2 elevation is supported by the prevention [28] or reversal [301] of the behavioral phenotype by mGluR2/3 antagonists. It is also supported by (2R, 6R) hydroxynorketamine’s rescue of the phenotype [39] since (2R, 6R) hydroxynorketamine acts principally through mGluR2-related mechanisms [302,303].

Late increased mGluR2 can also be seen as consistent with the hypothesis discussed above that chronic blast injury induces a hypoglutamatergic state. Presynaptic mGluR2/3 receptors act as autoreceptors that, when stimulated, inhibit glutamate release [288]. Thus, by analogy, increased mGluR2 levels would be predicted to be a marker of a hypoglutamatergic state. This analogy assumes that a pharmacological blockade has the same physiological consequences as reducing receptor overexpression. Support for this argument can be seen in our previous findings that treatment with mGluR2/3 antagonists reverses blast-related anxiety and fear-related behaviors as well as long-term recognition memory [28,288].

## 16. Future Studies

While we have some understanding of the neurochemical basis of late behavioral effects, much work remains to understand the early molecular evolution of the disease. Chiefly, a better understanding of the molecular evolution of changes at the NVU level and beyond is needed. Although some important clues concerning the role of inflammation in this model have come from regional transcriptomic studies conducted on whole-tissue RNA [40], the difficulty of examining single cell-type effects may be limiting. Indeed, single-nuclei RNA sequencing has revealed a heterogeneity to microglia and astrocytes not appreciated at the morphological level [267,304]. For example, one study identified a set of about 100 genes highly enriched in microglia that have been proposed to represent a microglial “tool set” for sensing the brain milieu [267]. Single-nuclei RNA sequencing has, in addition, identified five transcriptionally, distinct types of astrocytes, which differ in their expression of inflammatory mediators [304]. A recent single-nucleus, multi-region transcriptomic analysis of the brain vasculature in AD further demonstrated the power of single-cell approaches applied to the vasculature [305]. Future studies at the single-cell transcriptomic level seem warranted in our model of low-level BINT.

Finally, a proximate cause for the late neurobehavioral abnormalities seems to lie in the elevated expression of the group II metabotropic receptor mGluR2 [301]. However, the molecular basis for why mGluR2 elevation occurs and its relationship to an early vascular insult are unclear. In addition to developing strategies to address the late neurobehavioral consequences, strategies aimed at preventing or reversing vascular damage or modulating immune responses may improve chronic neuropsychiatric symptoms associated with BINT in humans.

## Figures and Tables

**Figure 1 ijms-25-01150-f001:**
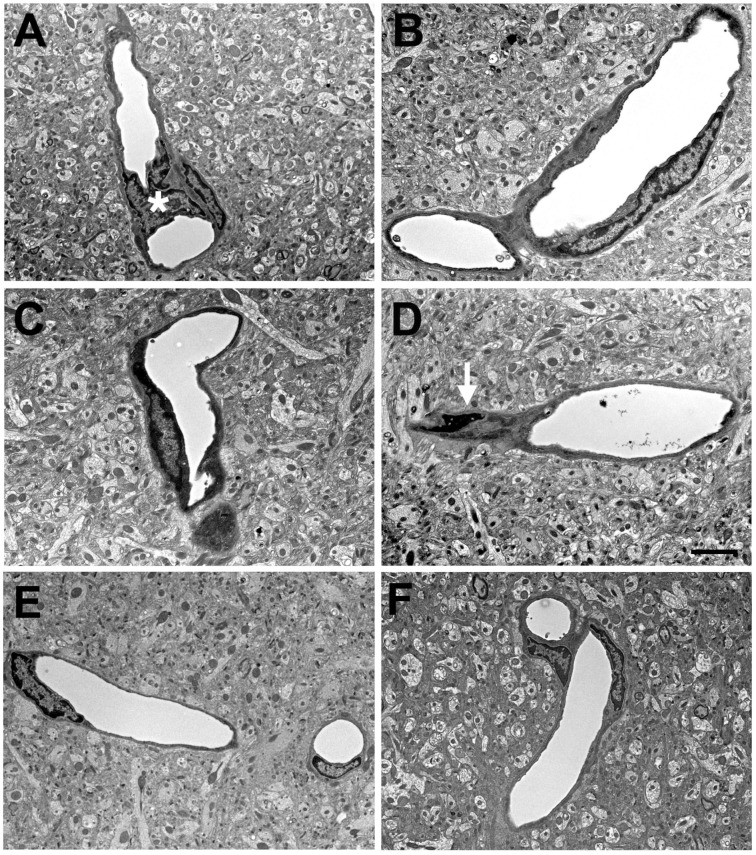
Blast-induced degenerative changes in cerebral microvessels. In panels (**A**–**D**), cerebral microvessels are shown from an animal that received a single 74.5 kPa blast exposure and was sacrificed 24 h later. Panels (**E**,**F**) illustrate longitudinally cut cerebral microvessels from non-blast exposed sham controls. All microvessels in panels (**A**–**D**) have lost their luminal circularity and the walls are irregular. In panel (**A**), a dysmorphic endothelial cell nucleus (*) is seen in the vessel lumen. In panel (**D**), the nucleus of a perivascular cell (arrow) with degenerative changes is indicated. Despite the destruction of the microvessels, the surrounding neuropil appears intact. Scale bar: 1 µm. Figure is reproduced from Gama Sosa et al. [34].

**Figure 2 ijms-25-01150-f002:**
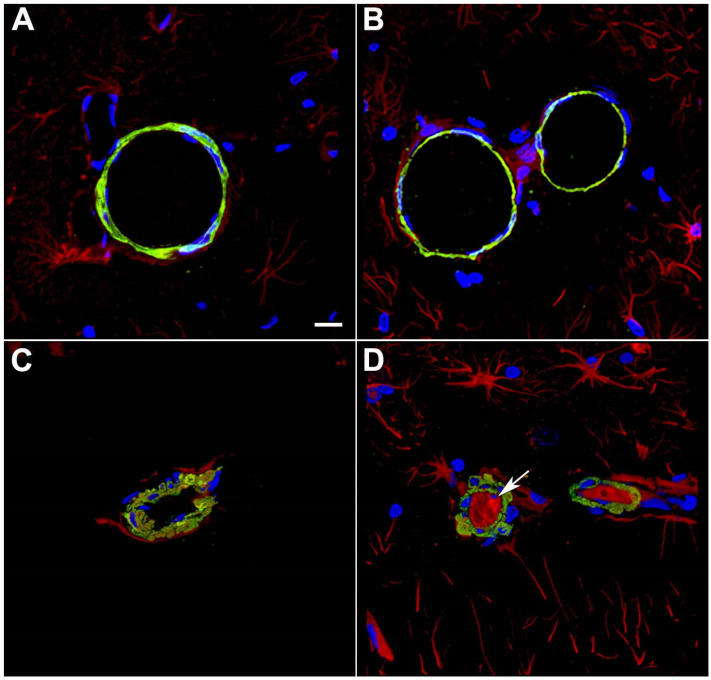
Disrupted smooth muscle layers and intraluminal astrocytic processes in blood vessels 10 months after blast exposure. Sections from rats euthanized 10 months after the last blast exposure were immunostained for GFAP (red) and α-SMA (green) with a DAPI nuclear counterstain (blue). Panels (**A**–**D**) show images from the hippocampal stratum lacunosum moleculare from control (**A**,**B**) or blast-exposed (**C**,**D**) rats. Note the irregularity and vacuolation in the smooth muscle apparent with α-SMA staining in the blast-exposed animal. An arrow in (**D**) points to intraluminal accumulation of GFAP. Scale bar, 50 μm. Figure is reproduced from Gama Sosa et al. [35].

**Figure 3 ijms-25-01150-f003:**
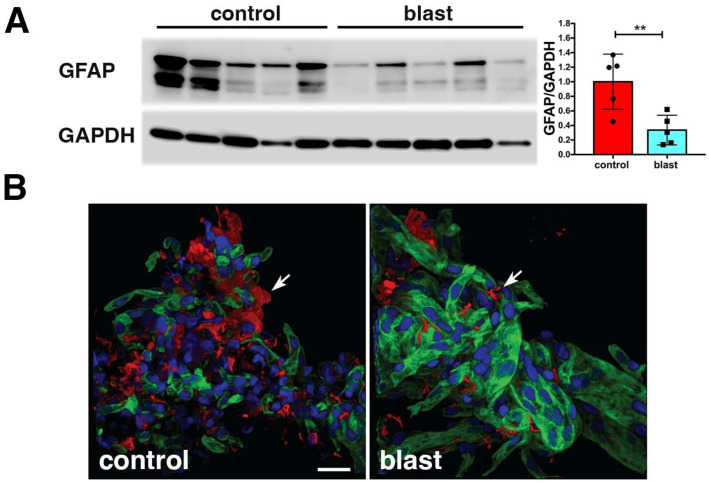
Reduced GFAP and fewer astroglial attachments in isolated brain vascular fractions following BINT. Panel (**A**) shows immunoblotting for GFAP (top panel) from five control and five blast-exposed brain-derived vascular fractions. Quantification of the blot with expression normalized to glyceraldehyde 3-phosphate dehydrogenase (GAPDH, lower panel) is shown on the right. All lanes were loaded with 10 μg of protein and contained protein from individual animals. Error bars indicate the SEM (** *p* < 0.01, unpaired *t*-test). Panel (**B**) shows isolated vascular fractions from control and blast-exposed brains immunostained for GFAP (red) and laminin (green) and counterstained with DAPI (blue). Arrows indicate GFAP-immunostained attachments to the isolated vasculature. Note reduced GFAP-labeled attachments in the blast-exposed vessels. All animals were euthanized 6 weeks after blast exposure. Scale bar, 25 μm. Panels (**A**,**B**) are from Gama Sosa et al. [35].

**Figure 4 ijms-25-01150-f004:**
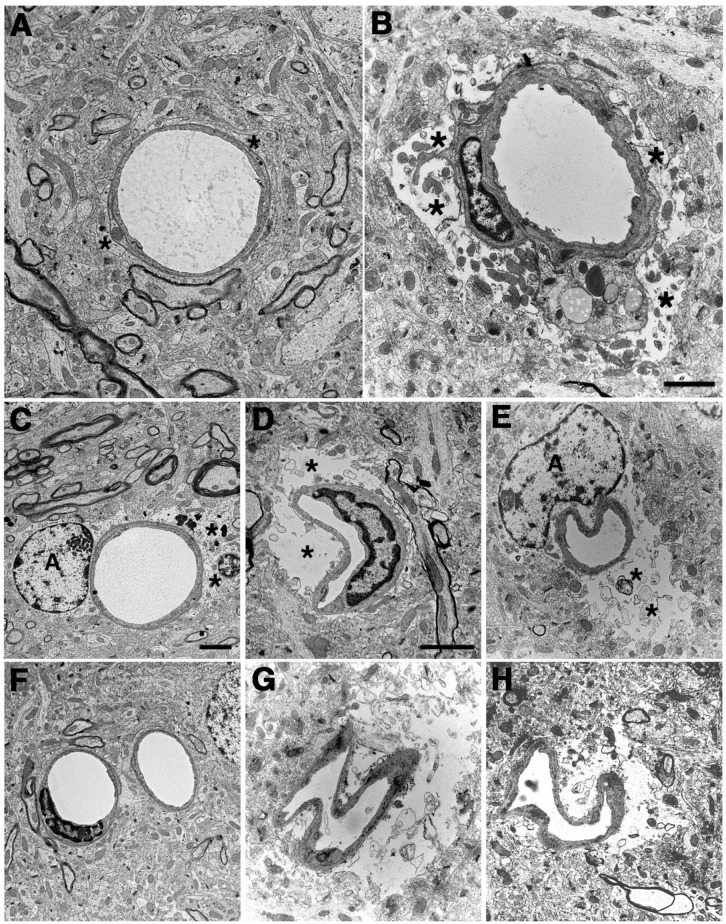
Blast-induced swelling and degeneration of astrocytic endfeet. Electron micrographs are shown from sections of the frontal motor cortex of control (**A**,**C**,**F**) and blast-exposed (**B**,**D**,**E**,**G**,**H**) rats. Animals were euthanized 6 weeks following 3 × 74.5 kPa blast exposures. Asterisks (*) indicate astrocytic endfeet, which are swollen and contain degenerating organelles in all blast-exposed microvessels. The lumens of microvessels in blast-exposed animals also appear irregular and collapsed. Perivascular astrocytes (labeled A) are visible in panels (**C**,**E**). Scale bars, 2 μm. Scale bar in panel (**B**) applies to panels (**A**,**B**). Scale bar in panel (**C**) applies to panels (**C**–**H**). Scale bar in panel (**D**) applies only to panel (**D**). Figure is reproduced from Gama Sosa et al. [35].

**Figure 5 ijms-25-01150-f005:**
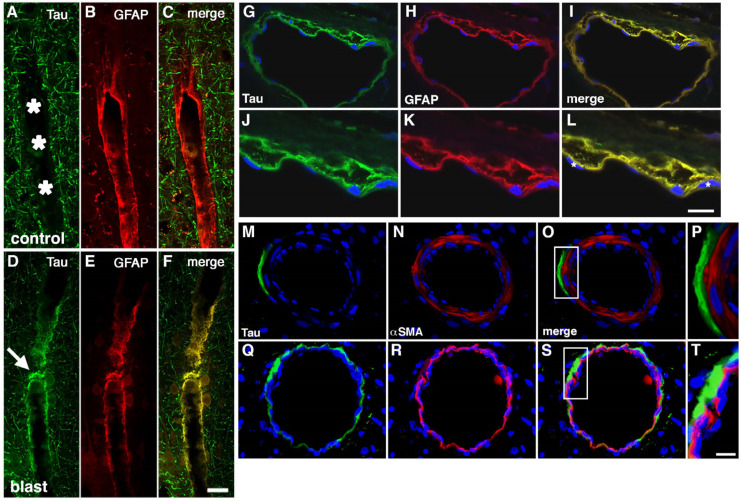
Perivascular p-tau in astroglial processes 10 months after blast exposure. Penetrating cortical vessels from control (**A**–**C**) and blast-exposed rats (**D**–**F**) immunostained for AT270 (green, **A**,**D**) and GFAP (red, **B**,**E**) are shown. Asterisks in panel (**A**) point to the vascular lumen. The arrow in panel (**D**) indicates p-tau staining. Scale bar in panel (**F**): 20 μm. In panels (**G**–**L**), a thalamic vessel is shown from a blast-exposed rat sacrificed 10 months after the last blast exposure immunostained with AT270 (green, **G**,**J**) and GFAP (red, **H**,**K**). Merged images are shown in panels (**I**,**L**). DAPI is shown in blue. Panels (**J**-**L**) show higher-power images of the vessel shown in panels (**G**–**I**). Note the localization of p-tau immunostaining mostly within GFAP immunostained perivascular astroglial processes. Asterisks in panel (**L**) indicate endothelial cell nuclei identifiable by their elongated appearance. Scale bar in panel (**L**): 40 μm for panels (**G**–**I**); 20 μm for panels (**J**–**L**). Panels (**M**–**T**) show a lack of p-tau in the vascular smooth muscle layer. Shown is a pial cortical vessel (**M**–**P**) or a thalamic vessel (**Q**–**T**) from a blast-exposed rat sacrificed 10 months after the last blast exposure. Sections were immunostained with AT270 (green, **M**,**Q**) and α-SMA (red, **N**,**R**). DAPI is shown in blue. Merged images are shown in panels (**O**,**S**). Panels (**P**,**T**) show higher-power images of the boxed areas indicated in panels (**O**,**S**). Note the lack of co-localization of p-tau immunostaining with the α-SMA stained smooth muscle layer. Scale bar is 40 μm for (**M**–**O**,**Q**–**S**); 10 μm for (**P**,**T**). Figures are reproduced from Dickstein et al. [33].

**Figure 6 ijms-25-01150-f006:**
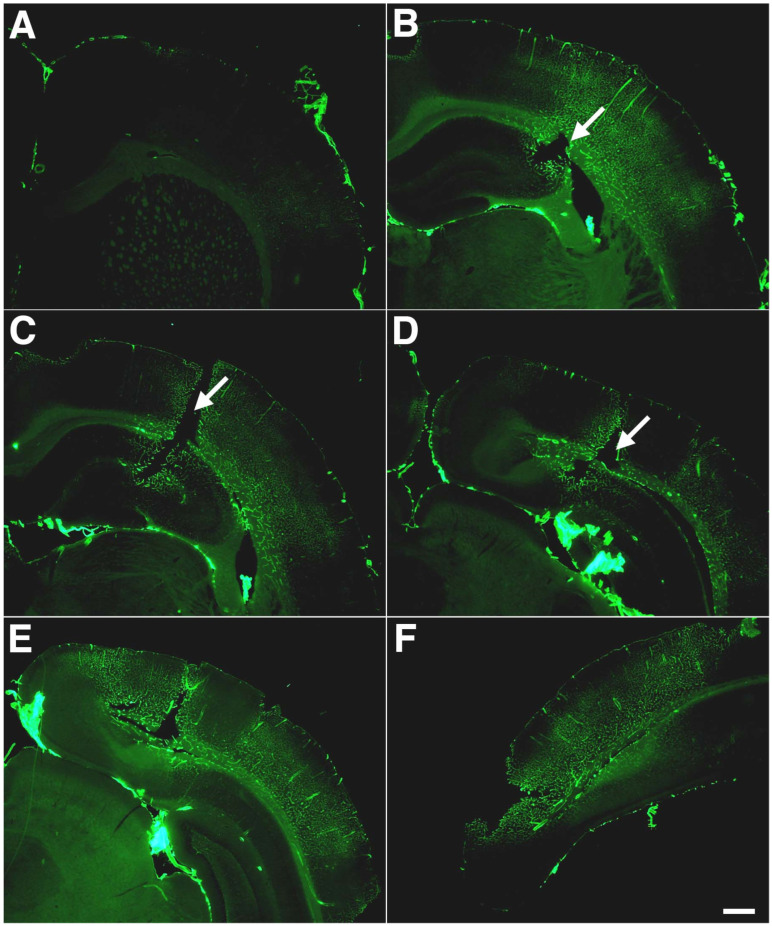
Altered extracellular matrix immunostaining of blast-exposed animals. Shown is collagen IV immunostaining without pepsin pretreatment of serial sections (**A**–**F**) taken 1200 μm apart from a blast-exposed rat that received 3 × 74.5 kPa exposures and was sacrificed 10 months after exposure. A focal cortical lesion is indicated by arrows in panels (**B**–**D**). Note the extensive lateral and rostro-caudal extent of altered collagen IV immunostaining without pepsin pretreatment. Scale bar: 750 μm. Figures are reproduced from Gama Sosa et al. [34].

**Figure 7 ijms-25-01150-f007:**
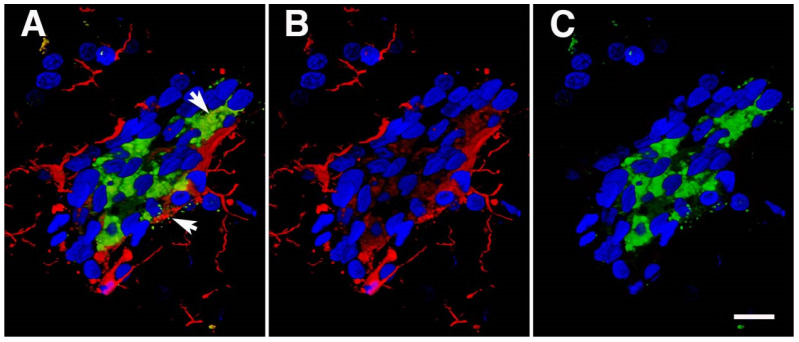
Activated perivascular microglia. Shown is a luminal view of a patch of perivascular activated M1 microglia expressing MHCII. Panel (**A**) shows merged images; panel (**B**), Iba1 immunostaining (microglia, red); and panel (**C**), MHCII immunostaining (activated M1 microglia, green). Arrows in panel (**A**) show an apoptotic-activated microglial cell (green) and a degenerating perivascular microglial cell (red). Scale, 20 μm. Figure is reproduced from Gama Sosa et al. [38].

**Figure 8 ijms-25-01150-f008:**
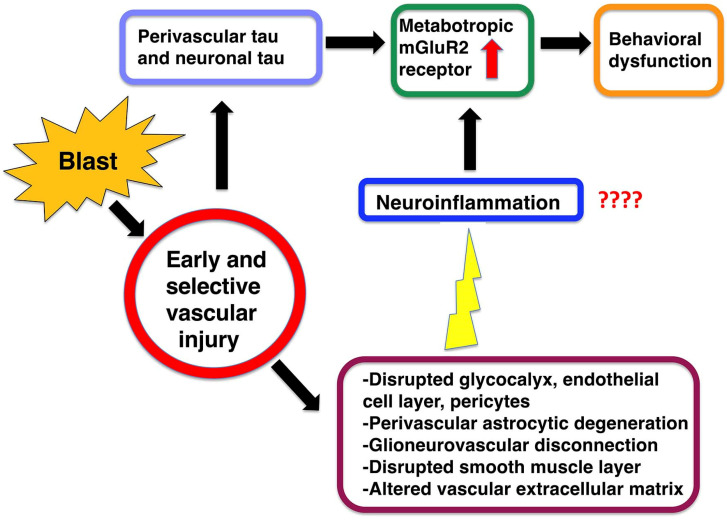
Hypothetical scenario for how early vascular injury leads to a delayed and chronic neurobehavioral phenotype. Details are discussed in the text.

**Table 1 ijms-25-01150-t001:** Rodent models of blast exposure utilizing overpressures of 80 kPa or less.

Species	Blast Exposure	Body Shiel-Ding	Head Restraint	Vascular Pathology	Non-Vascular Pathology	References
Mouse	Live explosives; 17.2 to 37.9 kPa	None	None	BBB permeability increase at 30 days but not at 7 days in 37.9 kPa-exposed animals based on T1 MRI contrast enhancement.	Increased fractional anisotropy on diffusion tensor imaging in thalamus and hypothalamus.	[83]
Mouse	Shock tube; 77 kPa	None	Yes and no	Dysmorphic capillaries; swollen, perivascular astrocytic endfeet thickened and often containing amorphous material and damage to capillary basal lamina.	p-tau CTE-like pathology; light and EM hippocampal neuronal pathology; ultrastructural damage to myelinated axons; and acute behavioral change prevented by head restraint.	[82]
Rat	Shock tube; single 70 kPa	No	Yes	None described.	At 5 days post injury, neuronal degeneration and increased density of inflammatory cells in frontal and parietal cortex, hippocampus, and thalamus.	[84]
Rat	Shock tube; single or repeated (5 × 70 kPa)	None	None	None described.	Microglial activation and reactive astrocytosis with 5×, but not single blast exposure at 35 days after exposure.	[85]
C57BL/6 mice	Live explosives; single-exposure 46.6 kPa peak overpressure; subjects prone; front of head exposure	None	None, but report no detectible motion of head or torso	Ultrastructural changes, including alterations in endothelial cells, pericytes, and basement membranes as well as astrocytic endfeet swelling.	Acute axonal damage seen by silver staining; ultrastructural abnormalities in myelin sheaths, mitochondria, and synapses; and proteomic changes in axonal and synaptic proteins.	[86,87,88,89]
C57BL/6 mice	Shock tube; single-exposure 25 kPa peak overpressure; subject prone outside of tube; front of head exposure	None	None	Multifocal areas of BBB breakdown as judged by Evans Blue and fluorescein isothiocyanate dextran extravasation 3 h to 7 days after exposure; clusters of reactive astrocytes and microglia at sites of BBB breakdown.	None described.	[90,91]
Rat	Shock tube; repetitive exposure (2×); 75.8 kPa; left side of head exposed	Body shielded	Head motion restricted	Tight junction (TJ) proteins (zonula occludens-1 [ZO-1], occludin, and claudin-5), decreased at 6 h and 24 h in males, but not in females; females decreased GFAP at 6 h; males decreased GFAP at 24 h. At 24 h, vascular integrity and astrocytic endfoot coverage of blood vessels decreased in males; by 7 d, no significant differences in TJ or GFAP levels between groups in either sex.	Regional impairments in synaptic mitochondrial respiration but not in mitochondrial respiration in glia.	[92,93]
Rat	Shock tube; single or multiple (up to 30) 27.5 to 58.6 kPa exposures; head on or side exposures	None	Yes	24 h, variable changes in occludin and claudin-5 depending on region and exposure.	Pathology not described.	[94]

**Table 2 ijms-25-01150-t002:** Pathophysiological events following blast injury and their relationship to later behavioral disturbance.

Component	Observations	Potential Functional Consequences	Relationship to Chronic Behavioral Phenotype
Endothelial cell	Acute endothelial cell damage, at least in part mediated by a thoracic effect if the blast involves a full-body exposure.	Altered BBB properties as a result of direct endothelial cell damage as well as disrupted endothelial cell connections to other mural cells, the vascular basement membrane, and perivascular astrocytes.	Predates
Endothelial glycocalyx	Glycocalyx degradation acutely.	Altered BBB regulation; greater leukocyte adhesion to endothelial cells and sensitivity of the endothelial cells to injury; impaired sensing of luminal shear forces.	Predates and appears to be repaired by the time the behavioral phenotype emerges.
Vascular smooth muscle	Changes in the microarchitecture of the smooth muscle layer; acute, delayed, and prolonged alterations in cerebrovascular reactivity.	Altered long-term vascular reactivity in the NVU.	Predates
Gliovascular connections	Decreased GFAP and lost astrocytic endfeet in isolated vascular fractions; swollen and dysmorphic astrocytic endfeet in vivo.	Altered BBB function, neurovascular coupling, perivascular glutamate clearance and calcium regulation; and immune cell entry.	Predates
Neurovascular connections	Decreased neuronal intermediate filaments (NF-L, NF-M, NF-H, and α-internexin) in isolated vascular fractions.	Impaired cerebral autoregulation and activity-dependent neurovascular coupling.	Predates
Glymphatics	Decreased brain Aβ 42.	Could imply increased glymphatic transport, although recent studies suggest that increased transendothelial transport out of the brain may also be a factor.	Predates
Perivascular tau	Perivascular p-tau in astrocytes.	Early vascular damage; seed for spread of neuronal tau later.	Predates
Pericytes	Pericytes and their normal association with other components of the NVU disrupted acutely.	Altered BBB maintenance, regulation of immune cell entry into CNS and neuroinflammatory states; disrupted neurovascular coupling.	Predates
Extracellular matrix	Chronic degradation and remodeling of the extracellular matrix.	Altered regulation of BBB permeability; chronic effects on perineuronal nets.	Predates
Perivascular inflammation	Activation of TNFα-regulated genes and downregulation of *P2ry12* between 6 weeks and 12 months after blast exposure; late perivascular inflammation with presence of activated microglial.	Perivascular neuroinflammatory environment leading to a hyperglutamatergic state early followed by a hypoglutamatergic one late; inflammation mediated neurodegeneration; and cytokine-mediated effects on behavioral function.	May correlate with appearance of the behavioral phenotype.
Neuronal mGlu2	Elevation late; behavioral phenotype reversed by mGluR2/3 antagonists.	Hypoglutamatergic synapse.	Correlates with appearance of the behavioral phenotype.

## Data Availability

Data underlying the basis of this report is available from the authors.

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
