# Peer review of "The Neurovascular Unit as a Locus of Injury in Low-Level Blast-Induced Neurotrauma"

_ijms, 2024, doi:10.3390/ijms25021150_

Round 1
Reviewer 1 Report
Comments and Suggestions for Authors
This review article addresses the question whether impairment of the neurovascular unit (NVU) plays a role in chronic neurobehavioral effects that follow low-level blast-induced neurotrauma (BINT). The authors have integrated their own findings with those from the literature of both, experimental animal models and clinical research and reports, mainly related to military injuries.
As compared to high-level blast induced neurotrauma, low level BINT is still relatively little studied, yet its incidence is rather high. This is a relevant topic to the field of Neurotrauma, in general, and specifically to that type of non-contact brain injury which is induced by low-level blast. The persistent of neurobehavioral changes that follow such injury requires better understanding of the mechanisms of these changes, and this comprehensive review covers these aspects.
In Figure 8, the authors present hypothetical scenario for how early vascular injury leads to a delayed and chronic neurobehavioral phenotype. Thus, based on their own and others reports they conclude that each of the factors they discussed in the review may potentially contribute to the resulting NVU pathology induced by low level blast injury.
There are more the 300 references most of which cover basic physiological pathways and how their function is impaired as a result of brain injury in general, with recent references related to BINT both in animal models and in the human military-related settings.
Author Response
We thank the reviewer for their generous comments and clear interest in our work. The only major change we have made to the manuscript is that in order to respond to reviewer #2’s request for a better summary of the individual sections we have added a second table which breaks each of the subareas down and summarizes their findings. We have also spell checked and made some minor copy edits to the manuscript.
Reviewer 2 Report
Comments and Suggestions for Authors
The authors present a comprehensive manuscript describing biological effects from scientific studies in both animals and humans following blast injury in the military. Overall, the study is well conceived, and a great deal of work was put into the research review in multiple areas related to brain physiology. In particular Figure 8 on page 27 is helpful to see the overall results. However, further clarification is needed. Although the authors provide a summary in Section 15-"Linking neurovascular and chronic neurobehavioral effects that follow BINT" and indicate how the findings may play a role in their vascular hypothesis for long term effects, it would be helpful if overall findings can be summarized with indications from each area studied and then identify important areas to consider for early injury findings and later findings following the injury.
Comments on the Quality of English LanguageSome edits needed.
Author Response
We thank the reviewer for their generous comments and clear interest in our work. We can appreciate the reviewer’s request for a better summary of the individual sections. Adding a summary at the end of each section, we suspect would not be that useful to a reader given the length of the article. Instead, we have added a table which summarizes the findings of each section (Table 2), their pathophysiological implications and relationship to the behavioral phenotype. This table has been inserted at the beginning of section 15. Together with Figure 8, we think it should give the reader a clear summary of the findings and the pathophysiological model being proposed. I have also redrafted section 15 somewhat, moving the paragraph on mGluR2 form the beginning of the section to the end which puts the story more in chronological order. I have also added some additional detail on why elevation of mGluR2 late is consistent with a late hypoglutamatergic state which is alluded to in section 15. I have also spell checked the document and made some minor copy edits to the manuscript here and there to correct grammatical errors and smooth out some awkward writing. Significant modifications to the text are indicate in blue in the revised manuscript. we hoe the conclusions of the manuscript are now more apparent.